# Nickel Salicylaldoxime-Based Coordination Polymer as a Cathode for Lithium-Ion Batteries

**Evgenii V. Beletskii [1], Daniil A. Lukyanov [1], Petr S. Vlasov [1], Andrei N. Yankin [2], Arslan B. Atangulov [1], Vladimir V. Sizov [1] and Oleg V. Levin [1,*]**

[1] Institute of Chemistry, St. Petersburg state University, 199034 St. Petersburg, Russia; belochkin@yandex.ru (E.V.B.); lda93@yandex.ru (D.A.L.); p.vlasov@spbu.ru (P.S.V.); lowen96@yandex.ru (A.B.A.); v.sizov@spbu.ru (V.V.S.)

[2] Department of Physics and Engineering, ITMO University, 197101 St. Petersburg, Russia; yankin_88@inbox.ru

[*] Correspondence: o.levin@spbu.ru; Tel.: +7-(812)-4286900

**Abstract:** Conjugated coordination polymers attract attention as materials for electrochemical energy storage, mostly as cathode materials for supercapacitors. Faradaic capacity may be introduced to such materials using redox-active building blocks, metals, or ligands. Using this strategy, a novel hybrid cathode material was developed based on a $Ni^{2+}$ metal-organic polymer. The proposed material, in addition to double-layer capacitance, shows high pseudocapacitance, which arises from the contributions of both the metal center and ligand. A tailoring strategy in the ligand design allows us to minimize the molecular weight of the ligand, which increases its gravimetric energy. According to computational results, the ligand makes the prevailing contribution to the pseudocapacitance of the material. Different approaches to metal–organic polymer (MOP) synthesis were implemented, and the obtained materials were examined by FTIR, Raman spectroscopy, powder XRD, SEM/EDX (energy-dispersive X-ray spectroscopy), TEM, and thermal analysis. Energy-storage performance was comparatively studied with cyclic voltammetry (CV) and galvanostatic charge–discharge (GCD). As a result, materials with an excellent discharge capacity were obtained, reaching the gravimetric energy density of common inorganic cathode materials.

**Keywords:** MOP; NiSalen; lithium-ion; supercapacitor; cathode material; nickel

## 1. Introduction

The development of electric transportation, mobile electronic devices, the Internet of Things network, and renewable energy sources require new types of energy-storage devices with increased values of both energy and power density [1]. Commercial lithium-ion batteries (LIBs), which dominate the current energy-storage market, deliver specific energies approaching 200 Wh kg$^{-1}$. However, the maximum specific power of traditional LIB metal oxide cathode materials is restricted by the diffusion-limited lithium ion intercalation and rarely exceeds 350 W kg$^{-1}$ [2,3]. By contrast, supercapacitors, with their fast ionic transport in soft matter and electrolyte instead of solid electrode materials, compensate for low specific energy of 10 Wh kg$^{-1}$ with extremely fast charge/discharge and high specific power reaching 10 kW kg$^{-1}$ [2,4]. Creation of an "ideal battery", which combines the energy of LIBs and the power of supercapacitors in one device, is an emerging target for current research. The bottleneck limiting the specific energy and power of whole LIB is the galvanostatic charge–discharge (GCD) performance of traditional oxide cathode materials, so the development of fast cathode materials is the first task to be solved in this way. In this course, hybrid materials combining redox- and double-layer capacitance are sought.



This task may be solved using metal–organic polymers (MOPs). A porous structure and high specific surface area, typical for MOPs, allow their implementation as electrode materials for supercapacitors [5–7]. MOPs based on nickel complexes of triphenylenehexamine (230 F $g^{-1}$) [8,9], terephthalic acid (1127 F $g^{-1}$) [10–13], and salicylic acid (1698 F $g^{-1}$) [14] are used as electrode materials in supercapacitors (Figure 1a). The use of redox-active MOPs provides additional pseudocapacitance, which increases the total capacity [15]. The redox transition may be provided by the metal center—for example, $Fe^{2+}/Fe^{3+}$ in the MOP with innocent ligands provides 75 mAh $g^{-1}$ of pseudocapacitance [16]. The introduction of redox-active ligands with a $Cu^{+}/Cu^{2+}$ transition achieves a twofold (147 mAh $g^{-1}$) increase in capacity [17].

Redox-active ligands for MOPs used as cathode materials should satisfy several requirements to fulfill the energy-storage task. The molar mass of the ligand per one electron should be reduced to increase the theoretical redox capacity. At the same time, the elevation of the redox potential of the ligand increases the specific energy of the battery. A special class of conductive polymers, polymeric bis (salicylideniminato) nickel complexes (*poly*NiSalens, Figure 1b) perfectly fits the above requirements [18]. It possesses high electronic conductivity [19], excellent cyclability as a cathode material for LIBs, and increased thermal stability. In addition, NiSalens are assembled from the simple synthetic blocks, 1,2-diamine and salicylaldehyde, using mild conditions, which offer the opportunities for extensive structural modifications and targeted fine-tuning of redox properties. The actual use of *poly*NiSalens, however, is hindered by synthetic problems. Thin films of NiSalen polymers are produced by electrochemical polymerization [18], but chemical oxidative polymerization is unable to produce such materials with sufficient quality in bulk [20].

**Figure 1.** (**a**) Nickel metal–organic polymers (MOPs) used as electrode materials for supercapacitors; (**b**) a representation of a NiSalen polymer.

Herein we report a nickel MOP inspired by the NiSalen structural motif. In contrast to the traditional NiSalen polymers, the proposed MOP was obtained by the polycoordination procedure, which is highly scalable. As a result, the material combining the advantages of both MOPs and *poly*NiSalens was synthesized. The performance of this MOP as a cathode material for LIB was examined by cyclic voltammetry (CV) and galvanostatic charge–discharge (GCD).

## 2. Materials and Methods

### 2.1. General Considerations

We purchased 1M $LiPF_6$ in 1:1:1 EC:DEC:DMC electrolyte, 1,4-dimethoxybenzene, HMTA hexamethylenetetramine, salicylic aldehyde, 1,2-diaminoethane. and anhydrous acetonitrile from Sigma-Aldrich. Nickel acetylacetonate obtained from local suppliers was sublimed in vacuo prior to use. Tetrahydrofuran (THF) obtained from local suppliers was stored over potassium hydroxide KOH, distilled over $LiAlH_4$, and then over sodium/benzophenone under Ar. Other chemicals were purchased from local suppliers and used as received. Solvents for MOP preparation were purified as described in literature prior to use [21]. The 2,5-dihydroxyterephthalaldehyde [22] and 2,5-dihydroxyterephthalaldehyde dioxime [23] were prepared as described previously. FTIR spectra

were recorded in KBr pellets on an IRAffinity-1 spectrometer. Raman spectra were obtained on a Bruker Senterra spectrometer using a 785 nm laser with 10 mW beam power. XRD was performed on a Bruker D2 Phaser diffractometer. SEM images were obtained on a Zeiss Merlin microscope. TEM was performed on a Zeiss Libra 200FE microscope. Thermal analysis was carried out at 5 K/min rate on a Netzsch TG 209F1 Libra apparatus in an Ar atmosphere.

### 2.2. Synthesis

MOP samples **I–IV** were obtained from the 2,5-dihydroxyterephthalaldehyde dioxime ligand by the following protocols:

**I.**  Obtained according to the previously described protocol in aqueous 1,4-dioxane with $Ni(OAc)_2$ as a nickel source and NaOAc to maintain the *p*H [24]. Yield 193.7 mg, 76.6%.

**II.**  Warm solutions of ligand (0.1962 g, 1 mmol) in glacial acetic acid (6 mL) and nickel acetate tetrahydrate (0.2490 g, 1 mmol) in the same solvent (4 mL) were mixed and stirred at 100 °C for 16 h. The brick-red precipitate was filtered, washed with acetic acid and 1,4-dioxane, and dried in vacuo at 100 °C for 4 h. Yield 166.3 mg, 65.8%.

**III.**  To the solution of ligand (0.1962 g, 1 mmol) in a mixture of dry acetonitrile (10 mL) and 1,4-dioxane (5 mL), the solution of nickel acetylacetonate (0.2569 g, 1 mmol) in acetonitrile (5 mL) was gradually introduced. The carrot-colored mixture was stirred in a sealed flask at 60 °C for 16 h. The product was filtered, washed thoroughly with acetonitrile, and dried in vacuo. Yield 178.0 mg, 70.4%.

**IV.**  To the solution of ligand (0.1962 g, 1 mmol) in freshly distilled water- and oxygen-free THF (10 mL), diisopropylethylamine (0.297 g 2.30 mmol) in THF (2 mL) was added under an argon atmosphere. The mixture turned deep-yellow and was stirred for 15 min. A solution of nickel acetylacetonate (0.2569 g, 1 mmol) in THF (8 mL) was added dropwise in 5 min, and the mixture was stirred for 1 h at r.t. Sienna-colored precipitate started to form. After being stirred at 60 °C for 16 h, the product was repeatedly washed with THF using centrifugation. Yield 219.7 mg, 86.9%. For further investigations, synthesis was reproduced without final drying. The "wet-clay" product obtained after centrifugation was resuspended in *N*-methylpyrrolidone (NMP) , and THF was removed from the suspension by stirring in vacuo at RT for 3 days, affording the ca. 10% dispersion of the product in NMP.

### 2.3. Half-Cell Fabrication

Slurry consisting of 20% of MOP, 70% of carbon black and 10% PVDF (*w/w*) was prepared by dry grinding in the mortar followed by the addition of NMP (10 mL per 1 g) and treatment in a blade homogenizer at 10 000 rpm. The resulting paste was applied on the 22 μm Al foil using a "Dr. Blade"-type applicator as a layer with a 200 μm thickness. The active mass thickness was 15 μm, and the average loading was 0.8 mg/cm$^2$. The resulting electrode was dried in vacuo at 80 °C for 12 h. Disks with d = 15.5 mm cut from the prepared electrode sheet were used as cathodes. CR2032 coin cells with a Li disk (d = 15.5 mm, h = 0.6 mm) as the anode, 1M $LiPF_6$ in 1:1:1 EC:DEC:DMC as the electrolyte, and Celgard® 2500 as the separator were assembled in a dry box filled with Ar.

### 2.4. Electrochemical Studies

Cyclic voltammograms were recorded on an Autolab PGStat 30 potentiostat in a coin cell with a lithium foil as the counter and a reference electrode. Potentials are hereinafter referenced as Li/Li$^+$. GCD experiments were performed on an Elins P20-X8 potentiostat in the 2.5–4.0 V range using the following sequence: 25 cycles at 1C, 3 cycles at 2C, 3 cycles at 5C, 3 cycles at 10C, and 25 cycles at 1C. GCD rates were calculated on the basis of 100 mAh g$^{-1}$ capacity.

*2.5. Computations*

Density functional theory (DFT) calculations were carried out with the long-range corrected CAM-B3LYP hybrid functional [25] and a 6-311+G* basis set for all atoms. Full geometry optimizations were performed for model MOPs containing up to three nickel atoms and up to three ligands; the resulting structures were verified by vibrational frequency analysis. All DFT calculations were carried out using the Gaussian 16 package [26].

## 3. Results and Discussion

First 2,5-dihydroxyterephthalic aldehyde was prepared and converted to a corresponding oxime according to the literature. From this oxime, four samples of MOP with $Ni^{2+}$, **I–IV**, were prepared under various conditions (Scheme 1). A few different methods were used in order to optimize the MOP preparation. Since the precipitation in the aqueous solution proceeded immediately during the mixing of the ligand and nickel acetate, one may expect an amorphous nature or low crystallinity of the product. In order to delay the fast precipitation, an attempt at synthesis in acetic acid was conducted to obtain sample **II**. In this case, the precipitation occurred with some delay indicating both lower deprotonated ligand concentration and higher solubility of intermediate complexes in this medium. Even though the product was thoroughly washed with 1,4-dioxane and dried in a vacuum, it could have retained undesirable traces of acetic acid. To exclude any acidic contamination, another source of nickel ions, sublimated nickel acetylacetonate, was used instead of nickel acetate. A mixture of aprotic solvents (1,4-dioxane-acetonitrile 1:1 v/v) was utilized for the preparation of **III**. The reaction proceeded with a delay and resulted in the desired precipitate, nevertheless the mother solution contained a residual soluble nickel complex. This may be explained by competition of the chelating ligands in the reversible reaction. To overcome this, the ligand was deprotonated with DIPEA in dry THF prior to the introduction of $[Ni_3(acac)_6]$, affording sample **IV**. Moreover, to prevent a conglomeration of **IV** particles, the procedure of an all-wet operation was developed, in which the THF was replaced with NMP without complete drying of the product.

**Scheme 1.** Synthetic route to MOPs **I–IV**.

FTIR and Raman spectra of the resulting materials were recorded. FTIR spectra (Figures S1–S4, see Supplementary Materials) of all samples show strong vibrations at 3384–3387 $cm^{-1}$ and 1640–1641 $cm^{-1}$ characteristic for stretching of O-H and C=N bonds, respectively. Signals in the fingerprint region are close in intensity and position for all samples. A few special features were also found. Sample **I** has a broad signal at 3200–3600 $cm^{-1}$, which indicates the presence of residual water, and vibration at 1556 $cm^{-1}$, which is probably caused by traces of acetate. Samples **II** and **III** demonstrate sharper signals, as compared to **I** and **IV**. The vibration at 1640 $cm^{-1}$ (C=N) in the sample **IV** has a shoulder at the high-energy side. This may be attributed to the influence of molecular arrangement or to the presence of other oxime species. Raman spectra of all samples (Figure S5) do not indicate noticeable differences and show bands at 1625 (C=N), 1496 (C-$C_{Ar}$), and 1284 (C-O) $cm^{-1}$. Sample **IV** demonstrates remarkably strong luminescence upon 785 nm Raman laser excitation. XRD diffractograms for all four samples (Figure S6) show high crystallinity. The diffraction patterns are virtually the same, but exact peak positions differ slightly, indicating the impact of the preparation protocol on the lattice parameters. In addition, the peaks are less pronounced in the case of **I**,

which indicates its lower crystallinity. The crystallite sizes of samples **I–IV** were calculated using the Scherrer equation, which results in 8–11 nm for **I**, 15–25 nm for **II**, 14–17 nm for **III**, and 16–17 nm for **IV**. The resulting MOP shows high thermal stability, with its onset temperature determined by TGA for sample **IV** as 322 °C. A significant mass loss (19.26%) appearing in the interval between 298 and 378 °C. constituted ca. 48 Da per repeating unit. The value was higher than expected for abstraction of two water molecules, indicating degradation of the material.

The morphologies and composition of electrode materials were examined using scanning electron microscopy (SEM, Figure 2a). The SEM image of sample **I** shows lumps of amorphous matter. Crystals were not observed even at high magnifications. In sample **II**, the pieces are smaller and demonstrate better arrangement, consistent with XRD data. Micrographs of **III** show small crystalline agglomerates. Crystallinity is more prominent in the case of **IV**, where the selenite rose-like, flake-shape agglomerates are clearly visible at high magnification. The agglomerates were 520 ± 70 nm in diameter and consist of plates with an average diameter of 160 ± 50 and 5 ± 1 nm thickness. The observed effect may be caused both by the low rate of polymer formation during the preparation and by the all-wet workup technique, which prevents the conglomeration. Transmission electron microscopy (TEM) also confirms the nanocrystalline (below 20 nm) type of material particles (Figure 2b,c).

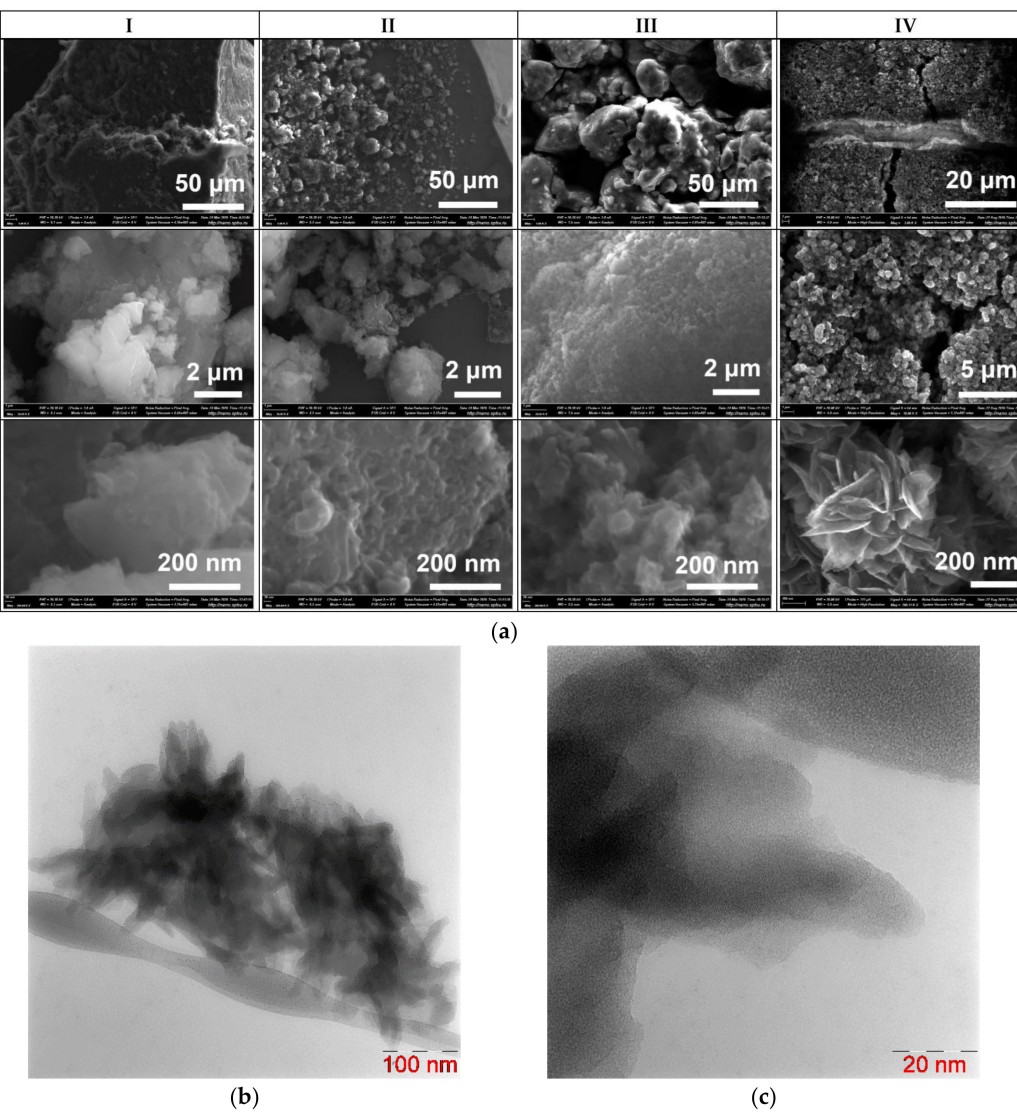

**Figure 2.** (**a**) SEM images of electrode materials I-IV, TEM images of **IV** at (**b**) lower magnification and (**c**) higher magnification.

A simultaneous elemental analysis of the samples during the SEM study was performed (Table 1). Unfortunately, nickel is the sole heavy element present in the material composition, and the real values for the C, O, and N content are often affected by the contamination of the residual atmosphere in the microscope chamber. However, according to the obtained results, sample **I** shows the best compliance with the idealized composition. Sample **IV** has a significant, but still acceptable deviation in elemental composition. By contrast, the relative nickel content in samples **II** and especially **III** is much lower than expected.

**Table 1.** EDX elemental analysis of samples **I–IV**.

| Atomic Ratio | C/Ni | N/Ni | O/Ni |
|:---:|:---:|:---:|:---:|
| Calculated | 8 | 2 | 4 |
| **I** | 8.0 ± 1.8 | 2.0 ± 0.6 | 4.5 ± 1.0 |
| **II** | 17.5 ± 4.4 | 4.5 ± 1.9 | 8.3 ±2.9 |
| **III** | 22.2 ± 3.5 | 7.9 ± 2.6 | 10.8 ± 2.6 |
| **IV** | 12.0 ± 0.3 | N/A [1] | 5.3 ± 0.3 |

[1] Not measured.

The CV of samples **I–IV** (Figure 3) demonstrates significant differences. Sample **I** exhibits two anodic peaks at 3.54 and 3.88 V, and corresponding cathodic peaks at 3.34 and 3.69 V, which are retained during cycling. Samples **II–IV** have only one pair of anodic and cathodic peaks at 3.81 and 3.67, 3.82 and 3.67, and 3.82 and 3.75 V, respectively. Samples **III** and **IV** exhibit much greater irreversible capacity than **I** and **II**. The CV curve of **IV** is markedly broadened, corresponding to a pseudocapacitive type of process and indicating the significant contribution of the double layer capacitance of a well-developed surface of nanocrystalline MOP.

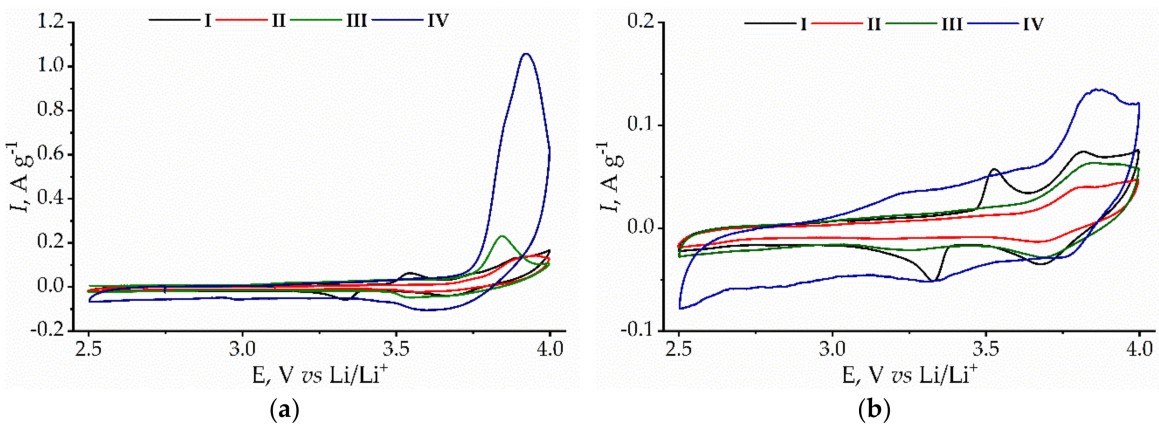

**Figure 3.** Cyclic voltammetry (CV) of **I–IV** (**a**) before galvanostatic charge–discharge (GCD); (**b**) after the last GCD cycle; rate 0.05 mV/s, 1M LiPF$_6$ in 1:1:1 EC:DEC:DMC.

Figure 4 demonstrates the cycle-to-cycle evolution of charge–discharge curves for the fabricated MOP lithium batteries at constant current densities of 1C, 2C, 5C, and 10C (1C = 100 mAh g$^{-1}$) and cut-off voltages of 4.0 and 2.5 V. The discharge of the MOPs starts from 3.8 V, which is 0.2 V higher than the value for *poly*NiSalen analog [18]. The first cycle discharge capacity of the MOPs reaches 102 mAh g$^{-1}$ for **I**, 95 mAh g$^{-1}$ for **II**, 130 mAh g$^{-1}$ for **III**, and 212 mAh g$^{-1}$ for **IV**. These values are more than one order of magnitude higher than for electrochemically polymerized *poly*NiSalen (10 mAh g$^{-1}$) and are comparable to commercial cathode materials for LIBs, such as LCO (140 mAh g$^{-1}$) or LFP (170 mAh g$^{-1}$) [27]. Even after 60 cycles of GCD, the capacity for new MOPs is still 5 times higher than for *poly*NiSalen. Such enhancement cannot be explained only by the lower molecular weight of the monomeric unit, thus suggesting higher practical oxidative doping levels of the materials and a contribution from the MOPs' double-layer capacitance.

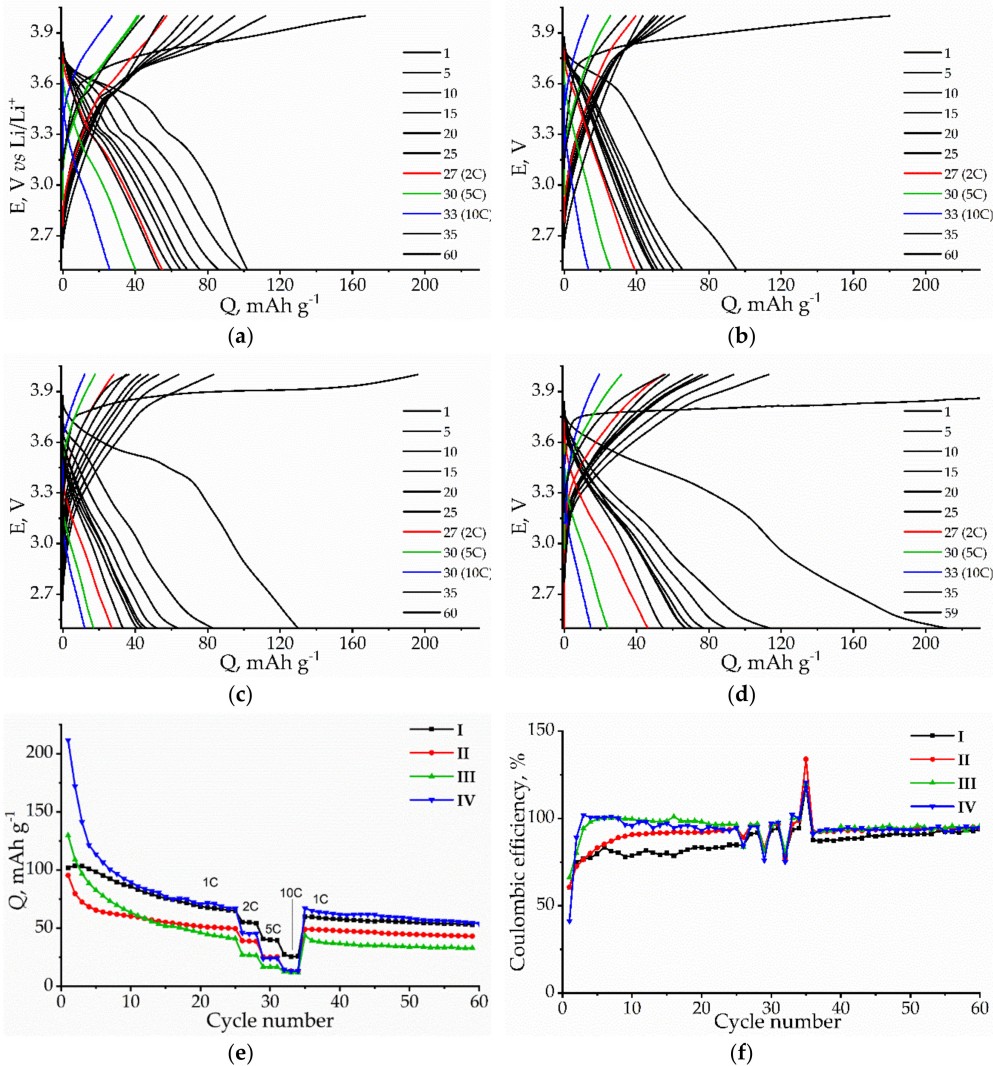

**Figure 4.** GCD curves for (**a**) **I**; (**b**) **II**; (**c**) **III**; (**d**) **IV**; (**e**) cycling performance and (**f**) coulombic efficiencies of all samples at 1C, 2C, 5C, and 10C in 1M LiPF$_6$ in a 1:1:1 EC:DEC:DMC Li anode.

However, all samples show a sharp decrease of the discharge capacity after the first cycles, stabilizing at the 15th cycle at values from ca. 80 mAh g$^{-1}$ for **I** and **IV** to ca. 50 mAh g$^{-1}$ for Ni-deficient **II** and **III**. Such behavior may be attributed to the rupture of the initial MOP material due to the ion intercalation upon the charge–discharge process and partial pulverization of the active material. After the completion of the initial capacity drop, the capacity of the materials is retained at high discharge rates, decreasing in **I**, the best case, from 69 mAh g$^{-1}$ at 1 C to 14 mAh g$^{-1}$ at 10C. The comparison of these samples demonstrates that the performance of the material depends significantly on the preparation method. As anticipated, sample **IV**, obtained in anhydrous conditions, shows the highest capacity at a 1C discharge rate. Surprisingly, elevated discharge rates highlight sample **I**, which retains a significant capacity even when being discharged at 10C.

Molecular-level insight into the structure and redox properties of the MOPs was provided by quantum chemical calculations for short-chain oligomers (Figure S8). In contrast to *poly*NiSalens [28], the structure of the reported MOP is perfectly planar, which can be attributed to the rigidity of the ligand and the coordination site, which is further enhanced by the formation of (N)O–H . . . O(C) hydrogen bonds between ligands. The stepwise increase of the total charge on oligomeric chains does not result in appreciable changes of the oxidation state of nickel atoms, suggesting a ligand-centered oxidation

mechanism and highlighting the key role of the non-innocent ligand in the pseudocapacitance of the material (Figure S9). These results are in line with that reported previously for *poly*NiSalens [28,29].

Thus, a tailoring strategy was proposed, which allows for targeted development of novel metal–organic cathode materials for lithium-ion batteries. The nickel-containing conjugated coordination polymer was obtained in this study via implementation of this strategy on the basis of the rational revision of *poly*NiSalen materials from the energy-storage viewpoint. The results of the present study clearly demonstrate the advantages of nanostructured coordination polymers with the π–d conjugation chain. The synthesized materials display high capacities in excess of 200 mAh g$^{-1}$ and residual capacities of more than 50 mAh g$^{-1}$ after 60 cycles. The significant drop of the capacity upon GCD may be attributed to the rupture of the structure due to ion intercalation, which can be prevented by morphology control. The reported MOP material is an attractive object of further study in the development of electrochemical energy-storage devices, namely, sample **IV** as a prominent candidate in terms of specific energy and sample **I** in terms of specific power and charge–discharge stability.

**Supplementary Materials:** The following are available online at http://www.mdpi.com/1996-1073/13/10/2480/s1, Figures S1–S4: FTIR spectra, Figure S5: Raman spectra, Figure S6: XRD patterns for samples **I–IV**, Figure S7: TGA curve for sample **IV**, Figure S8—optimized geometries for dimeric and trimeric subchains, Figure S9—charge distribution diagram upon the stepwise oxidation.

**Author Contributions:** Conceptualization, D.A.L. and O.V.L.; synthesis, D.A.L., A.N.Y., and P.S.V.; electrochemistry E.V.B. and O.V.L.; computational studies, A.B.A. and V.V.S. All authors have read and agreed to the published version of the manuscript.

**Funding:** This research was funded by Russian Foundation for Basic Research (RFBR), grant number 18-29-04058.

**Acknowledgments:** We thank the Educational Resource Center of Chemistry, the Research Center for Magnetic Resonance, the Center for Chemical Analysis and Materials Research, the Center for X-ray Diffraction Studies, the Thermogravimetric and Calorimetric Research Centre, the Centre for Optical and Laser Materials Research, the Nanotechnology Center, and the Computing Centre of Saint Petersburg State University Research Park for the measurements and calculations provided.

**Conflicts of Interest:** The authors declare no conflict of interest.

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
