# Peer review of "Nickel Salicylaldoxime-Based Coordination Polymer as a Cathode for Lithium-Ion Batteries"

_energies, doi:10.3390/en13102480_

Round 1

Reviewer 1 Report

This manuscript reports on the synthesis and characterization of a new nickel-based metal-organic polymer for battery applications. Designing new materials for energy storage is of utmost fundamental and technological importance and this work proposes a promising, novel compound, which is clearly of interest. The research context is properly set in the introduction and the need for improved nickel-containing active materials is clearly demonstrated. Different preparation protocols lead to materials with different degrees of crystallinity and a correlation is demonstrated with the electrochemical response, with the most crystalline sample showing the best performances. This work is innovative and interesting and the manuscript clearly deserves to be published. Only a couple of minor points should be corrected/improved:

  • The text on the TGA measurements seems contradictory (page 4) : “Resulting MOP shows high thermal stability, its onset temperature was determined by TGA for the sample IV as 322°C. The value shows moderate temperature stability of the complexes.” Do the authors consider that the compounds have high or moderate thermal stability ? This should be clarified.
  • It is doubtful that conventional SEM, as used here, gives information on the molecular-scale arrangement, as claimed on page 5. Simply, the morphology is consistent with the presence of crystalline materials, especially for samples III and IV. The text should be corrected.
  • The compositional information drawn from EDX measurements is of a very limited interest. Clearly, XPS should be used to obtain relevant information on the composition. Either XPS data should be added or the paragraph on EDX should simply be removed.
  • The contribution of quantum-chemical modeling is disappointingly short and hasty. A clearer explanation, supported by a figure showing the change in charge density evolution upon electron removal, should be given to substantiate the sentence on page 7: “The stepwise increase of the total charge on oligomeric chains does not result in appreciable changes of the oxidations state of nickel atoms, suggesting a ligand-centered oxidation mechanism and highlighting the key role of the non-innocent ligand in the pseudocapacity of the material.”

I therefore suggest acceptance of this manuscript for publication in Energies when those minor points have been addressed.

Author Response

Comments and Suggestions for Authors

Response

Page and line numbers are given for “Track changes, All Markup” mode.

1

The text on the TGA measurements seems contradictory (page 4): “Resulting MOP shows high thermal stability, its onset temperature was determined by TGA for the sample IV as 322°C. The value shows moderate temperature stability of the complexes.” Do the authors consider that the compounds have high or moderate thermal stability? This should be clarified.

Thermal stability is high if compared to organic or metal-organic compounds and moderate in comparison to inorganic materials. The term “moderate” was removed to avoid the contradiction and the text was corrected accordingly (P4L162).

2

It is doubtful that conventional SEM, as used here, gives information on the molecular-scale arrangement, as claimed on page 5. Simply, the morphology is consistent with the presence of crystalline materials, especially for samples III and IV. The text should be corrected.

The term “molecular” was removed (P5L171).

3

The compositional information drawn from EDX measurements is of a very limited interest. Clearly, XPS should be used to obtain relevant information on the composition. Either XPS data should be added or the paragraph on EDX should simply be removed.

We agree that EDX data are not as representative as XPS. Unfortunately, we are unable to perform XPS now due to COVID-19. However, we believe that EDX data are better than nothing.

4

The contribution of quantum-chemical modeling is disappointingly short and hasty. A clearer explanation, supported by a figure showing the change in charge density evolution upon electron removal, should be given to substantiate the sentence on page 7: “The stepwise increase of the total charge on oligomeric chains does not result in appreciable changes of the oxidations state of nickel atoms, suggesting a ligand-centered oxidation mechanism and highlighting the key role of the non-innocent ligand in the pseudocapacity of the material.”

The required figure was added to the ESI (Figure S9). The reference to this figure was added to the main text (P7L228, P8L244).

Technical corrections

Acknowledgements were corrected (P8L253)

Author’s name was changed in manuscript (P1L4) and in ESI.

Reviewer 2 Report

In this study, the application of nickel salicylaldoxime based coordination polymer as a cathode for lithium-ion batteries. The authors showed the nickel-containing conjugated coordination polymer can be the energy storage viewpoint. The coordination polymers with the π-d conjugation demonstrate the advantages and display high capacities and residual capacities. The work is potentially suitable for publication. I recommend the paper is accepted with some minor changes.

  1. What different for I to IV in the nanostructure?
  2. “Molecular-level insight into the structure and redox properties of the MOPs was provided by quantum chemical calculations for short-chain oligomers” if changing the long-chain oligomers, the capacities can increasing or decreasing?
  3. Can the authors explain the advantages of Salen type ligands in the energy storage devices?

Reviewer 3 Report

In the manuscript from Levin and co-worker, NiII-based metal-organic conjugated polymers were synthesized as cathode material for Li ion battery application. The material was carefully characterized. And the energy storage performance was studied by CV and GCD. The material showed high capacity. Even the cycling performance is not very good, I think this study is good to publish in the Energies.

Author Response

Comments and Suggestions for Authors

Response

The authors appreciate greatly the review of the manuscript. As there is no point to answer, no response is provided

Technical corrections

Acknowledgements were corrected (P8L253)

Author’s name was changed in manuscript (P1L4) and in ESI.

Reviewer 4 Report

The manuscript "Nickel salicylaldoxime based coordination polymer as a cathode for lithium-ion batteries" report the synthesis and characterization of conjugated coordination polymers to be used in lithium-ion batteries. The reported research is of interest to the scientific community and can be considered for publication in Journal Energies, not before clarifying some issues.

  1. Did the authors calculate interplanar distance and unit cell parameter from XRD? Or crystallite size? It will be interesting to see the evolution of thease parameters from one synthesis to other.
  2. Did the authors check the type crystalline phase in synthesized materials? Or determined the crystallinity? The crystal simmetry?
  3. What was the reason for providing the TGA for the IV sample? What about the TGA of the other samples? Do they look different or have the same trend?
  4. The name of Table 1. is "SEM images and EDX patterns of electrode materials I-IV.", but in the table could not be found any EDX pattern. It seems the authors missed them.
  5. In conclusions, could the authors make a recommendation which one of the synthesized compound will exhibit more efficiency in the lithium batteries?

Reviewer 5 Report

  1. Being a more generally accepted term, I suggest to change "pseudocapacity" to "psuedocapacitance".
  2. In the abstract, "gravimetric energy" should be "gravimetric energy density".
  3. The abbreviation GCD is never defined and is not a standard term in the field.
  4. p. 1, line 38: "capacity" is used instead of "capacitance".
  5. Since all testing is done in a half-cell configuration using Li metal as a counter electrode, is it relevant to talk about pseudocapacitance for the material? Since the counter electrode will undergo redox reactions, all redox reactions at the working electrode will be matched by redox reactions at the working electrode (as opposed to in a capacitor-type cell setup).
  6. The statement "A few other methods were used in order to improve the MOP preparation" on p. 3 does not make sense since no initial method is described.
  7. It is stated that the Raman spectra in Figure S5 do not show appreaciable differences. What about the strong peak to the far left for sample IV, for example?
  8. Can you explain the origin of the strong luminescence for sample IV?
  9. XRD peaks are said to be less pronounced for sample I, but they are similarly unpronounced (or even more so) for sample IV.
  10. How can you infer the molecular arrangement from SEM, as stated on p. 5?
  11. "Micrographs" should be used instead of "microphotographs".
  12. Why was the N/Ni atomic ratio not measured for sample IV? It doesn't seem to make sense to leave this blank for this particular sample.
  13. What is the coulombic efficiency in the cycling tests? What is the reversibility of the redox reaction?
  14. Please provide a reference for the claim that the discharge voltage is 0.2 V higher than for the polyNiSalen analog (p. 6). Can you elaborate on why the value is higher for your system?
  15. Please avoid showing different C-rates in the same cycling curve graph (Figure 4a-d).
  16. Can you elaborate further on why the different preparation methods give such different cycling results?
  17. Please explain further how the capacity drop can be prevented by morphology control, since this is not demonstrated in the paper.
